# A Variational Perspective on High-Resolution ODEs

**Hoomaan Maskan**
Umeå University
hoomaan.maskan@umu.se

**Konstantinos C. Zygalakis**
University of Edinburgh
k.zygalakis@ed.ac.uk

**Alp Yurtsever**
Umeå University
alp.yurtsever@umu.se

## Abstract

We consider unconstrained minimization of smooth convex functions. We propose a novel variational perspective using forced Euler-Lagrange equation that allows for studying high-resolution ODEs. Through this, we obtain a faster convergence rate for gradient norm minimization using Nesterov's accelerated gradient method. Additionally, we show that Nesterov's method can be interpreted as a rate-matching discretization of an appropriately chosen high-resolution ODE. Finally, using the results from the new variational perspective, we propose a stochastic method for noisy gradients. Several numerical experiments compare and illustrate our stochastic algorithm with state of the art methods.

## 1 Introduction

Smooth convex minimization is a fundamental class in optimization with broad applications and a rich theoretical foundation that enables the development of powerful methods. In fact, the theory and methods developed for other problem classes, such as non-smooth convex or smooth non-convex optimization, often build upon the work done in smooth convex optimization. As a result, the study of smooth convex minimization has a significant impact on the broader field of optimization. Over the last two decades, first-order methods –those relying solely on gradient information, in contrast to methods like Newton's that require Hessian information– have seen a lot of interest both in theory and applications due to their efficiency and adaptability for large-scale data-driven applications. Among these, gradient descent stands as one of the simplest and oldest. When equipped with an appropriate step-size, gradient descent guarantees a suboptimality gap (objective residual) of order $\mathcal{O}(1/k)$ after $k$ iterations.

In his seminal work, Nesterov [1983] has shown that the gradient method can achieve faster rates by incorporating momentum deviations. Nesterov's accelerated gradient algorithm (NAG) ensures a convergence rate of $\mathcal{O}(1/k^2)$ which is an order of magnitude faster than the gradient descent. Remarkably, this rate matches information-theoretical lower bounds for first-order oracle complexity, meaning that NAG is optimal and no other first-order method can guarantee an essentially faster convergence rate [Nesterov, 2003].

The original proof of Nesterov [1983], known as the *estimate sequence technique*, is a highly algebraic and complex procedure, difficult to interpret, and provides arguably limited insight into why momentum deviations help with the convergence rates [Hu and Lessard, 2017]. Therefore, many researchers have tried to provide a better understanding of momentum-based acceleration through different perspectives. For example, Su et al. [2016], Shi et al. [2021], and Sanz Serna and Zygalakis [2021] consider a *continuous time perspective*; Lessard et al. [2016] and Fazlyab et al. [2018] use integral quadratic constraints and control systems with non-linear feedbacks; Muehlebach and Jordan [2019, 2022, 2023] present a dynamical perspective; Attouch et al. [2020, 2021] utilize inertial dynamic involving both viscous damping and Hessian-driven damping; Zhu and Orecchia [2014] view acceleration as a linear coupling of gradient descent and mirror descent updates; and Ahn and Sra [2022] provide an understanding of the NAG algorithm through an approximation of the proximal point method. Our work is focused on the *continuous-time perspective*.

37th Conference on Neural Information Processing Systems (NeurIPS 2023).

In [Su et al., 2016], the authors derive a second-order ordinary differential equation (ODE) that exhibits trajectories similar to those of the NAG algorithm in the limit of an infinitesimal step-size $s \to 0$. This result has inspired researchers to analyze various ODEs and discretization schemes to gain a better understanding of the acceleration phenomenon. Notably, Wibisono et al. [2016] demonstrate that the ODE presented in [Su et al., 2016] is a special case of a broader family of ODEs that extend beyond Euclidean space. They achieve this by minimizing the action on a Lagrangian that captures the properties of the problem template, an approach known as the *variational perspective*, and they discretize their general ODE using the *rate-matching* technique. In a related vein, Shi et al. [2021] proposed substituting the low-resolution ODEs (LR-ODEs) introduced in [Su et al., 2016] with high-resolution ODEs (HR-ODEs) which can capture trajectories of NAG more precisely.

Our main contribution in this paper is a novel extension of the variational perspective for HR-ODEs. A direct combination of these two frameworks is challenging, as it remains unclear how the Lagrangian should be modified to recover HR-ODEs. To address this problem, we propose an alternative approach that preserves the Lagrangian but extends the variational perspective. More specifically, instead of relying on the conventional Euler-Lagrange equation, we leverage the forced Euler-Lagrange equation that incorporates external forces acting on the system. By representing the damped time derivative of the potential function gradients as an external force, our proposed variational perspective allows us to reconstruct various HR-ODEs through specific damping parameters. More details are provided in Section 2.

Other contributions of our paper are as follows: In Section 3, we show that our proposed variational analysis yields a special representation of NAG leading to superior convergence rates than [Shi et al., 2019] in terms of gradient norm minimization. In Section 4, we propose an HR-ODE based on the rate-matching technique. We demonstrate that NAG can be interpreted as an approximation of the rate-matching technique applied to a specific ODE. Then, in Section 5, we extend our analysis to a stochastic setting with noisy gradients, introducing a stochastic method that guarantees convergence rates of $\tilde{\mathcal{O}}(1/k^{1/2})$ for the expected objective residual and $\tilde{\mathcal{O}}(1/k^{3/4})$ for the expected squared norm of the gradient. Finally, in Section 6, we present numerical experiments to demonstrate the empirical performance of our proposed method and to validate our theoretical findings.

**Problem template and notation.** We consider a generic unconstrained smooth convex minimization template:

$$\min_{x \in \mathbb{R}^n} f(x) \tag{1}$$

where $f : \mathbb{R}^n \to \mathbb{R}$ is convex and $L$-smooth, meaning that its gradient is Lipschitz continuous:

$$\|\nabla f(x) - \nabla f(y)\| \leq L\|x - y\|, \quad \forall x, y \in \mathbb{R}^n, \tag{2}$$

with $\|\cdot\|$ denoting the Euclidean norm. We denote the class of $L$-smooth convex functions by $\mathcal{F}_L$.

Throughout, we assume that the solution set for Problem (1) is non-empty, and we denote an arbitrary solution by $x^*$, hence $f^* := f(x^*) \leq f(x)$ for all $x \in \mathbb{R}^n$.

**The NAG Algorithm.** Given an initial state $x_0 = y_0 \in \mathbb{R}^n$ and a step-size parameter $s > 0$, the NAG algorithm updates the variables $x_k$ and $y_k$ iteratively as follows:

$$\begin{aligned} y_{k+1} &= x_k - s\nabla f(x_k), \\ x_{k+1} &= y_{k+1} + \frac{k}{k+3}(y_{k+1} - y_k). \end{aligned} \tag{NAG}$$

**Low-Resolution and High-Resolution ODEs.** Throughout, by LR-ODE, we refer to the second-order ODE introduced and studied in [Su et al., 2016], which exhibits a behavior reminiscent of momentum-based accelerated methods like Heavy Ball (HB) and NAG in the limit of step-size $s \to 0$. Unfortunately, this generic ODE cannot distinguish the differences between HB and NAG. It is important to note, however, that the guarantees of HB and NAG differ significantly in discrete time. Specifically, HB has guarantees for only a specific subset of Problem (1), whereas NAG guarantees a solution for all instances of (1) with appropriate step sizes. Therefore, it's worth noting that the LR-ODE may not fully capture certain important aspects of the NAG trajectory. Conversely, by HR-ODEs, we refer to the ODEs proposed in [Shi et al., 2021], which extend the LR-ODE by incorporating gradient correction terms. Through this extension, HR-ODEs provide a more precise representation of the trajectory for these distinct algorithms.

## 2 External Forces and High-Resolution ODEs

Consider the Lagrangian

$$\mathcal{L}(X_t, \dot{X}_t, t) = e^{\alpha_t + \gamma_t}\left(\frac{1}{2}\|e^{-\alpha_t}\dot{X}_t\|^2 - e^{\beta_t}f(X_t)\right). \tag{3}$$

Here, $\dot{X}_t \in \mathbb{R}^d$ is the first time-derivative of $X(t)$, and $\alpha_t, \beta_t, \gamma_t : \mathbb{T} \to \mathbb{R}$ are continuously differentiable functions of time that correspond to the weighting of velocity, the potential function $f$, and the overall damping, respectively. Using variational calculus, we define the action for the curves $\{X_t : t \in \mathbb{R}\}$ as the functional $\mathcal{A}(X) = \int_{\mathbb{R}} \mathcal{L}(X_t, \dot{X}_t, t)dt$. In the absence of external forces, a curve is a stationary point for the problem of minimizing the action $\mathcal{A}(X)$ *if and only if* it satisfies the Euler Lagrange equation $\frac{d}{dt}\{\frac{\partial \mathcal{L}}{\partial \dot{X}_t}(X_t, \dot{X}_t, t)\} = \frac{\partial \mathcal{L}}{\partial X_t}(X_t, \dot{X}_t, t)$. This was used in [Wibisono et al., 2016, Wilson et al., 2021] to calculate the LR-ODEs for convex and strongly convex functions[1]. Note that the Euler-Lagrange equation as written, does not account for an external force, $F$ (which is non-conservative). In this case, the Euler-Lagrange equation should be modified to the forced Euler-Lagrange Equation

$$\frac{d}{dt}\left\{\frac{\partial \mathcal{L}}{\partial \dot{X}_t}(X_t, \dot{X}_t, t)\right\} - \frac{\partial \mathcal{L}}{\partial X_t}(X_t, \dot{X}_t, t) = F, \tag{4}$$

which itself is the result of integration by parts of Lagrange d'Alembert principle [Campos et al., 2021]. Using the Lagrangian (3) we have

$$\frac{\partial \mathcal{L}}{\partial \dot{X}_t}(X_t, \dot{X}_t, t) = e^{\gamma_t}(e^{-\alpha_t}\dot{X}_t), \quad \frac{\partial \mathcal{L}}{\partial X_t}(X_t, \dot{X}_t, t) = -e^{\gamma_t + \alpha_t + \beta_t}(\nabla f(X_t)). \tag{5}$$

Substituting (5) in (4) gives

$$\ddot{X}_t + (\dot{\gamma}_t - \dot{\alpha}_t)\dot{X}_t + e^{2\alpha_t + \beta_t}\nabla f(X_t) = e^{\alpha_t - \gamma_t}F. \tag{6}$$

In what follows, we will present various choices of the external force $F$, including two for convex functions and one for strongly convex functions.

### 2.1 Convex Functions

**First choice for convex functions.** Let us first consider the following external force:

$$F = -\sqrt{s}e^{\gamma_t}\frac{d}{dt}[e^{-\alpha_t}\nabla f(X)]. \tag{7}$$

In this case, (6) becomes

$$\ddot{X}_t + (\dot{\gamma}_t - \dot{\alpha}_t)\dot{X}_t + e^{2\alpha_t + \beta_t}\nabla f(X_t) = -\sqrt{s}e^{\alpha_t}\frac{d}{dt}\left[e^{-\alpha_t}\nabla f(X_t)\right]. \tag{8}$$

It is possible to show the convergence of $X_t$ to $x^*$ and establish a convergence rate for this as shown in the following theorem (proof in Appendix A.1). The proof of this theorem (and the subsequent theorems in this section) is based on the construction of a suitable Lyapunov function for the corresponding ODE (*e.g.* see [Siegel, 2019, Shi et al., 2019, Attouch et al., 2020, 2021]). This non-negative function attains zero only at the stationary solution of the corresponding ODE and decreases along the trajectory of the ODE [Khalil, 2002]. For this theorem (and the subsequent theorems), we will define a proper Lyapunov function and prove sufficient decrease of the function $f$ along the corresponding ODE trajectory.

**Theorem 2.1.** *Under the ideal scaling conditions $\dot{\beta}_t \leq e^{\alpha_t}$ and $\dot{\gamma}_t = e^{\alpha_t}$, $X_t$ in (8) will satisfy*

$$f(X_t) - f(x^*) \leq \mathcal{O}(e^{-\beta_t})$$

*for $f \in \mathcal{F}_L$.*

---

[1]Wilson et al. [2021] use different Lagrangian for strongly convex functions, but the methodology is the same.

Now, choosing parameters as

$$\alpha_t = \log(n(t)), \quad \beta_t = \log(q(t)/n(t)), \quad \dot{\gamma}_t = e^{\alpha_t} = n(t), \tag{9}$$

in (8) gives

$$\begin{cases} \ddot{X}_t + (n(t) - \frac{\dot{n}(t)}{n(t)} + \sqrt{s}\nabla^2 f(X_t))\dot{X}_t + (n(t)q(t) - \sqrt{s}\frac{\dot{n}(t)}{n(t)})\nabla f(X_t) = 0, \\ F = -\sqrt{s}e^{\gamma_t}\frac{d}{dt}[e^{-\alpha_t}\nabla f(X)], \end{cases} \tag{10}$$

which reduces to

$$\ddot{X}_t + \left(\frac{p+1}{t} + \sqrt{s}\nabla^2 f(X_t)\right)\dot{X}_t + \left(Cp^2 t^{p-2} + \frac{\sqrt{s}}{t}\right)\nabla f(X_t) = 0, \tag{11}$$

by taking $n(t) = \frac{p}{t}, q(t) = Cpt^{p-1}$.

*Remark* 2.1.1. For $p = 2, C = 1/4$, equation (11) corresponds to the (H-ODE) in [Laborde and Oberman, 2020].

**Second choice for convex functions.** Now, we consider an external force given by

$$F = -\sqrt{s}e^{\gamma_t - \beta_t}\frac{d}{dt}\left[e^{-(\alpha_t - \beta_t)}\nabla f(X_t)\right] \tag{12}$$

In this case, replacing $F$ in (6) gives

$$\ddot{X}_t + (\dot{\gamma}_t - \dot{\alpha}_t)\dot{X}_t + e^{2\alpha_t + \beta_t}\nabla f = -\sqrt{s}e^{\alpha_t - \beta_t}\frac{d}{dt}[e^{-(\alpha_t - \beta_t)}\nabla f(X_t)]. \tag{13}$$

We establish the following convergence result, and the proof can be found in Appendix A.2.

**Theorem 2.2.** *Under the modified ideal scaling conditions* $\dot{\beta}_t \leq e^{\alpha_t}, \dot{\gamma}_t = e^{\alpha t}, \ddot{\beta}_t \leq e^{\alpha_t}\dot{\beta}_t + 2\dot{\alpha}_t\dot{\beta}_t,$ $X_t$ *in (13) will satisfy*

$$f(X_t) - f(x^*) \leq \mathcal{O}\left(\frac{1}{e^{\beta_t} + \sqrt{s}e^{-2\alpha_t}\dot{\beta}_t}\right),$$

*for* $f \in \mathcal{F}_L$.

Taking the same parameters as in (9) gives

$$\begin{cases} \ddot{X}_t + (n(t) - \frac{\dot{n}(t)}{n(t)} + \sqrt{s}\nabla^2 f(X_t))\dot{X}_t + (n(t)q(t) - \sqrt{s}(\frac{\dot{n}(t)}{n(t)} - \frac{\dot{q}(t)n(t) - \dot{n}(t)q(t)}{n(t)q(t)}))\nabla f(X_t) = 0, \\ F = -\sqrt{s}e^{\gamma_t - \beta_t}\frac{d}{dt}\left[e^{-(\alpha_t - \beta_t)}\nabla f(X_t)\right]. \end{cases} \tag{14}$$

which reduces to

$$\ddot{X}_t + \left(\frac{p+1}{t} + \sqrt{s}\nabla^2 f(X_t)\right)\dot{X}_t + \left(Cp^2 t^{p-2} + \frac{\sqrt{s}(p+1)}{t}\right)\nabla f(X_t) = 0, \tag{15}$$

for $n(t) = p/t, q(t) = Cpt^{p-1}$.

*Remark* 2.2.1. Note that setting $C = 1/4, p = 2$ will lead to the ODE

$$\ddot{X}_t + \left(\frac{3}{t} + \sqrt{s}\nabla^2 f(X_t)\right)\dot{X}_t + \left(1 + \frac{3\sqrt{s}}{t}\right)\nabla f(X_t) = 0. \tag{16}$$

This ODE was discretized using the Semi-Implicit Euler (SIE) and the Implicit Euler (IE) discretization schemes in [Shi et al., 2019]. The corresponding optimization algorithms were shown to accelerate. In addition, note that the convergence rate proved in Theorem 2.2 is faster than its counterpart in Theorem 2.1.

## 2.2 Strongly Convex Functions

Our analysis is applicable to strongly convex functions as well. Consider the Lagrangian proposed in [Wilson et al., 2021] for strongly convex functions

$$\mathcal{L}(X_t, \dot{X}_t, t) = e^{\alpha_t + \beta_t + \gamma_t}\left(\frac{\mu}{2}\|e^{-\alpha_t}\dot{X}_t\|^2 - f(X_t)\right). \tag{17}$$

Then, the forced Euler-Lagrange equation (4) becomes

$$\ddot{X} + (-\dot{\alpha}_t + \dot{\gamma}_t + \dot{\beta}_t)\dot{X} + \frac{1}{\mu}e^{2\alpha_t}\nabla f(X) = \frac{F}{\mu e^{-\alpha_t + \gamma_t + \beta_t}}. \tag{18}$$

Taking $F = -\sqrt{s}e^{\alpha_t + \gamma_t}\frac{d}{dt}(e^{\beta_t}\nabla f(X_t))$ in (18) gives

$$\ddot{X} + (-\dot{\alpha}_t + \dot{\gamma}_t + \dot{\beta}_t)\dot{X} + \frac{1}{\mu}e^{2\alpha_t}\nabla f(X) = \frac{-\sqrt{s}e^{2\alpha_t - \beta_t}\frac{d}{dt}(e^{\beta_t}\nabla f(X_t))}{\mu}. \tag{19}$$

We can establish the following convergence result for $X_t$ in (19) to the unique minimizer $x^*$. The proof of this result is deferred to Appendix A.3.

**Theorem 2.3.** *Under the modified ideal scaling conditions $\alpha_t = \alpha$, $\dot{\beta}_t \leq e^{\alpha_t}$, $\dot{\gamma}_t = e^{\alpha_t}$, and $\dot{\beta}_t \geq 0$ $X_t$ in (19) satisfies*

$$f(X_t) - f(x^*) \leq \mathcal{O}(e^{-\beta_t}) \tag{20}$$

*for $\mu$-strongly convex function $f$.*

*Remark* 2.3.1. Taking $\alpha = \log(\sqrt{\mu})$ and $\gamma_t = \beta_t = \sqrt{\mu}t$ in (19) gives the NAG's corresponding HR-ODE

$$\ddot{X}_t + (2\sqrt{\mu} + \sqrt{s}\nabla^2 f(X_t))\dot{X}_t + (1 + \sqrt{\mu s})\nabla f(X_t) = 0, \tag{21}$$

for $\mu$-strongly convex function $f$ as in [Shi et al., 2021].

# 3 Gradient Norm Minimization of NAG

One of the implications of our variational study on HR-ODEs in Section 2 was the ODE (14). Reformulating this ODE gives

$$\begin{cases} \dot{X}_t = n(t)(V_t - X_t) - \sqrt{s}\nabla f(X_t) \\ \dot{V}_t = -q(t)\nabla f(X_t) - \sqrt{s}\frac{\dot{q}(t)n(t) - \dot{n}(t)q(t)}{n^2(t)q(t)}\nabla f(X_t). \end{cases} \tag{22}$$

Applying the SIE on (22) for $X(t) \approx X(t_k), V(t) \approx V(t_k), n(t_k) = p/t_k, q(t_k) = Cpt_k^{p-1}$, $p = 2, t_k = k\sqrt{s}$ and $C = 1/4$ gives

$$\begin{cases} x_{k+1} = x_k + \frac{2}{k}(v_k - x_{k+1}) - s\nabla f(x_k), \\ v_{k+1} = v_k - \frac{1}{2}(ks)\nabla f(x_{k+1}) - s\nabla f(x_{k+1}), \end{cases} \tag{23}$$

which is exactly the NAG algorithm. The interpretation of the NAG method as the SIE discretization of (22) has not been discussed before in the literature (see [Ahn and Sra, 2022] for the four most studied representations). It is precisely this connection with the ODE (22) though that inspires our choice of the Lyapunov function which in turn gives rise to a faster convergence rate. The following theorem formulates this result. The proof is in Appendix A.4 and it is based on the discrete Lyapunov analysis of (23). Similar convergence rate was very recently found by [Chen et al., 2022] through *implicit velocity* perspective on HR-ODEs which uses a different Lyapunov analysis than this work.

**Theorem 3.1.** *Consider the update (23). Then, if $f \in \mathcal{F}_L$ we have*

$$\min_{0 \leq i \leq k-1} \|\nabla f(x_i)\|^2 \leq \frac{12}{k^3 s^2}\|x_0 - x^*\|^2,$$

*and*

$$f(x_k) - f(x^*) \leq \frac{2}{sk(k+2)}\|x_0 - x^*\|^2$$

*for $0 \leq s \leq 1/L$, $v_0 = x_0$, and any $x_0 \in \mathbb{R}^n$.*

*Remark* 3.1.1 (Comparison with state of the art). The rate in Theorem 3.1 is improved compared to the previous rate found in [Shi et al., 2021], which is

$$\min_{0 \leq i \leq k} \|\nabla f(x_i)\|^2 \leq \frac{8568}{(k+1)^3 s^2}\|x_0 - x^*\|^2,$$

for $0 < s \leq 1/(3L)$ and $k \geq 0$.

# 4 Rate-Matching Approximates the NAG Algorithm

The ODE (11) when $p = 2, C = 1/4$ is equivalent to

$$\begin{cases} \dot{X}_t = \frac{2}{t}(Z_t - X_t) - \sqrt{s}\nabla f(X_t), \\ \dot{Z}_t = -\frac{t}{2}\nabla f(X_t). \end{cases} \tag{24}$$

which can be viewed as a perturbation of the LR-ODE

$$\begin{cases} \dot{X}_t = \frac{2}{t}(Z_t - X_t), \\ \dot{Z}_t = -\frac{t}{2}\nabla f(X_t). \end{cases} \tag{25}$$

We now show that when the rate-matching technique in [Wibisono et al., 2016] is applied to (25), the final algorithm reveals similar behavior as (24). This result is then used to approximately recover the NAG method using rate-matching discretization.

Applying the rate-matching discretization on the ODE (25) gives

$$\begin{cases} x_{k+1} = \frac{2}{k+2}z_k + \frac{k}{k+2}y_k, \\ y_k = x_k - s\nabla f(x_k), \\ z_k = z_{k-1} - \frac{1}{2}sk\nabla f(y_k). \end{cases} \tag{26}$$

which has a convergence rate of $\mathcal{O}(1/(sk^2))$ [Wibisono et al., 2016]. In the following proposition, we study the behavior of (26) in limit of $s \to 0$. The proof is given in Appendix A.5.

**Proposition 4.1.** *The continuous-time behavior of (26) is approximately*

$$\ddot{X}_t + \left(\frac{3}{t} + \sqrt{s}\nabla^2 f(X_t)\right)\dot{X}_t + \left(1 + \frac{\sqrt{s}}{t}\right)\nabla f(X_t) = 0, \tag{27}$$

*which is the the high-resolution ODE (11).*

The ODE (27) is the same as (24). In this sense, rate-matching implicitly perturbs the LR-ODE. The question that naturally arises is that when do we recover the HR-ODE (16) (which corresponds to the NAG algortihm through the SIE discretization) from the rate-matching technique? To answer, we will first perturb the LR-ODE (25) in the second line. Then, the rate-matching discretization is applied. Perturbing (25) gives

$$\begin{cases} \dot{X}_t = \frac{2}{t}(Z_t - X_t), \\ \dot{Z}_t = -\frac{t}{2}\nabla f(X_t) - \sqrt{s}\nabla f(X_t). \end{cases} \tag{28}$$

Discretizing (28) using the rate-matching method with $t_k = k\sqrt{s}$ gives

$$\begin{cases} x_{k+1} = \frac{2}{k+2}z_k + \frac{k}{k+2}y_k, \\ y_k = x_k - s\nabla f(x_k), \\ z_k = z_{k-1} - \frac{s}{2}(k+2)\nabla f(y_k), \end{cases} \tag{29}$$

which is extremely close to the NAG algorithm. Indeed, replacing $\nabla f(y_k)$ with $\nabla f(x_k)$ in the third line of (29) gives exactly the NAG method. Typically, $x_k$ and $y_k$ are very close. This is due to $x_k$ and $y_k$ having a difference of order $s$. Since in continuous time $X(t_k) \approx Y(t_k)$ (due to $s \to 0$), the HR-ODE of (29) is (16). This means that the corresponding HR-ODE of (29) is

$$\begin{cases} \dot{X}_t = \frac{2}{t}(Z_t - X_t) - \sqrt{s}\nabla f(X_t), \\ \dot{Z}_t = -\frac{t}{2}\nabla f(X_t) - \sqrt{s}\nabla f(X_t). \end{cases} \tag{30}$$

which is the perturbed version of (28) and the HR-ODE associated with the NAG algorithm.

# 5 Stochastic Extensions

In this section, we propose a stochastic variation of (23). We model noisy gradients by adding i.i.d noise $e_k$ with variance $\sigma^2$ to the gradients. Consider the update

$$\begin{cases} x_{k+1} = x_k + \frac{2s_k}{t_k}(v_k - x_{k+1}) - \frac{\beta s_k}{\sqrt{L}}(\nabla f(x_k) + e_k), \\ v_{k+1} = v_k - \frac{1}{2}(t_k s_k + \frac{2s_k\beta}{\sqrt{L}})(\nabla f(x_{k+1}) + e_{k+1}) \end{cases} \tag{31}$$

with $\beta \geq 2$. This update reduces to (23) when $e_k = 0, s_k = \sqrt{s} = \beta/\sqrt{L}, t_k = k\sqrt{s}$. We will refer to (31) as the Noisy NAG (NNAG) algorithm. NNAG is interesting due to its capability of dealing with perturbed gradients. This is the case in practical methods *e.g.* SGD [Bottou, 2010], SAG [Schmidt et al., 2017], SAGA [Defazio et al., 2014], SVRG [Johnson and Zhang, 2013], and etc. The following convergence result holds for NNAG, and its proof is in Appendix A.6.

**Theorem 5.1.** *Suppose $f \in \mathcal{F}_L$ and consider the NNAG method detailed in (31) with the following parameter choices:*

$$\beta \geq 2, \quad s_k = \frac{c}{k^\alpha}, \quad \text{and} \quad t_k = \sum_{i=1}^{k} s_i \quad \text{for some} \quad c \leq \frac{1}{\sqrt{L}} \quad \text{and} \quad \frac{3}{4} \leq \alpha < 1. \quad (32)$$

*We define the critical iteration $k_0$ as the smallest positive integer that satisfies*

$$k_0 \geq \left( \frac{\beta}{\frac{1}{c\sqrt{L}} + \frac{c\sqrt{L}}{8} \left( \sum_{i=1}^{k_0} \frac{1}{i^\alpha} \right)^2} \right)^{1/\alpha}. \quad (33)$$

*Then, the following bounds hold for all $k \geq k_0$ :*

$$\mathbb{E}[f(x_k)] - f(x^*) \leq \frac{\mathbb{E}[\varepsilon(k_0)] + \frac{\sigma^2 c^4}{(1-\alpha)^2} \left[ k_0^{3-4\alpha} - k^{3-4\alpha} \right] + \frac{\sigma^2 c^3 \beta}{2\sqrt{L}(1-\alpha)(3\alpha-2)} \left[ k_0^{2-3\alpha} - k^{2-3\alpha} \right] + \frac{\beta^2 c^2 \sigma^2}{2L(2\alpha-1)} \left[ k_0^{1-2\alpha} - k^{1-2\alpha} \right]}{\frac{c^2}{4(1-\alpha)^2} \left( (k^{1-\alpha}-1)^2 \right) + \frac{c\beta}{2\sqrt{L}(1-\alpha)} \left( k^{(1-\alpha)} - 1 \right)}$$

*if $\alpha > 3/4$, and*

$$\mathbb{E}[f(x_k)] - f(x^*) \leq \frac{\mathbb{E}[\varepsilon(k_0)] + 2\sigma^2 c^4 \left[ \log(\frac{k}{k_0}) \right] + \frac{8\sigma^2 c^3 \beta}{\sqrt{L}} \left[ k_0^{-1/4} - k^{-1/4} \right] + \frac{\beta^2 c^2 \sigma^2}{L} \left[ k_0^{-1/2} - k^{-1/2} \right]}{4c^2 \left( (k^{1/4}-1)^2 \right) + \frac{2c\beta}{\sqrt{L}} \left( k^{1/4} - 1 \right)} \quad (34)$$

*if $\alpha = 3/4$ with $\varepsilon(k) = \left( \frac{t_k^2}{4} + \frac{t_k \beta}{2\sqrt{L}} \right) (f(x_k) - f(x^*)) + \frac{1}{2} \| v_k - x^* \|^2$.*

Next, we show that slight modifications to the NNAG method gives rise to another stochastic method with a similar convergence rate as the NNAG algorithm, but more transparent proof (see Appendix A.7). This proof results in a convergence rate for $\mathbb{E}\left[ \min_{0 \leq i \leq k-1} \| \nabla f(x_i) \|^2 \right]$ with a rate of $O(\log(k)/k^{(3/4)})$. It remains a future work to show similar result for the NNAG update.

**Theorem 5.2.** *Suppose $f \in \mathcal{F}_L$ and consider the following modification of the NNAG method*

$$\begin{cases} x_{k+1} = x_k + \frac{2s_k}{t_k}(v_k - x_{k+1}) - \frac{s_k}{\sqrt{L}}(\nabla f(x_k) + e_k), \\ v_{k+1} = v_k - \frac{1}{2}((t_k)s_k)(\nabla f(x_{k+1}) + e_{k+1}) - s_k^2(\nabla f(x_{k+1}) + e_{k+1}). \end{cases} \quad (35)$$

*with the same parameter choices as in (32). Then, the following convergence bounds hold:*

$$\mathbb{E}[f(x_k)] - f(x^*) \leq \begin{cases} \frac{\mathbb{E}[\varepsilon(0)] + \frac{c^4 \sigma^2}{8} [16(1+\log(k))+32+6]}{2c^2 \left[ 2(k^{\frac{1}{4}}-1)^2 + k^{-\frac{3}{4}}(k^{\frac{1}{4}}-1) \right]} & \alpha = \frac{3}{4} \\ \frac{\mathbb{E}[\varepsilon(0)] + \frac{c^4 \sigma^2}{8} \left[ \frac{(4\alpha-2)}{(1-\alpha)^2(4\alpha-3)} + \frac{4(4\alpha-1)}{(1-\alpha)(4\alpha-2)} + \frac{4(4\alpha)}{(4\alpha-1)} \right]}{\frac{c^2}{2(1-\alpha)} \left[ \frac{(k^{1-\alpha}-1)^2}{2(1-\alpha)} + k^{-\alpha}(k^{(1-\alpha)}-1) \right]} & 1 > \alpha > \frac{3}{4} \end{cases}, \quad (36)$$

*with $\mathbb{E}[\varepsilon(0)] = \frac{1}{2} \| v_0 - x^* \|^2$. In addition, for $\alpha = 3/4$ we have*

$$\mathbb{E}\left[ \min_{0 \leq i \leq k-1} \| \nabla f(x_i) \|^2 \right] \leq \frac{2\sqrt{L}\mathbb{E}[\varepsilon(0)] + (2c^4 \sigma^2 \sqrt{L})(2\log(k) + 6 + \frac{3}{4})}{16c^3 \left( \frac{k^{3/4}-1}{3} + k^{1/4} - \frac{3}{2} + k^{1/2} \right)}. \quad (37)$$

*Remark* 5.2.1. The algorithm in (35) reduces to (23) when $e_k = 0, s_k = \sqrt{s} = 1/\sqrt{L}, t_k = k\sqrt{s}$.

*Remark* 5.2.2 (Connection to NAG). When there is no noise ($\sigma = 0$) we can also allow parameter $\alpha$ to be zero. This is because we do not need to alleviate the effect of the noise with a decreasing stepsize. Therefore, we recover the convergence rate of $O(1/k^2)$ for the NAG method when $c = 1/\sqrt{L}$.

*Remark* 5.2.3 (Comparison with [Laborde and Oberman, 2020]). Laborde et al., proposed a stochastic method with noisy gradients. Their method uses another presentation of the NAG algorithm (the presentation from EE discretization). Our rate (36) has the same order of convergence as [Laborde and Oberman, 2020]. However, their analysis did not achieve the bound (37) (see [Laborde and Oberman, 2020] Appendix C.4).

*Remark* 5.2.4 (Comparison between Theorems 5.1 and 5.2). The rate in (34) is asymptotically similar to (36). However, the transient behavior of (34) is faster than both (36) and the rate in [Laborde and Oberman, 2020] when $L$ is large (see Figure 1 top row). This is due to the tuning parameter $\beta$ which is usually set to $L$ or higher. This scenario (Large $L$) often happens in practice, *e.g.* in training a two-layer Convolutional Neural Network (CNN) [Shi et al., 2022].

*Remark* 5.2.5 (Limitations and future directions). One limitation of our theoretical analysis is that our convergence result for NNAG holds only for a large enough number of iterations $k \geq k_0$. However, in our numerical experiments we observed that the same bounds hold also for $k \leq k_0$, so we believe our result can be improved. Additionally, our proposed forces are currently defined only in Euclidean space, and we see potential for extending the framework to non-Euclidean spaces.

# 6    Numerical Results

In this section, we present our empirical results, divided into three parts: Theoretical upper bounds, binary classification with logistic regression, and classification with neural networks.

**Upper Bounds.**    First, we compare the bounds in (34) and (36) with the Proposition 4.5 in [Laborde and Oberman, 2020]. The results are depicted in Figure 1. In practical scenarios where $L$ is large [Shi et al., 2022] the bound (34) is lower than the other two for large enough iterations. This observation has motivated us to analyze the behavior of NNAG in practical scenarios such as binary classification and CNN training tasks.

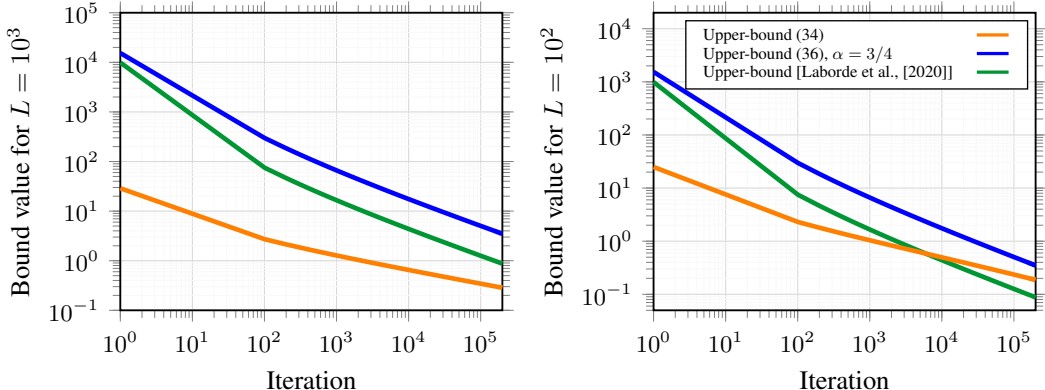

Figure 1: Comparison of our upper bounds with the state-of-the-art.

**Binary Classification.**    For this task we considered $d = 1000$ randomly generated samples of dimension $n = 10$ and labels. The problem can be written as

$$\min_{x \in \mathbb{R}^n} \quad \frac{1}{d} \sum_{i=1}^{d} \log(1 + e^{-y_i \langle x_i, x \rangle}), \tag{38}$$

where $y_i$ and $x_i$ denote the $i$th label and sample. For comparison, we considered the perturbed gradient descent method with Gaussian noise and decreasing step-size $s_k = 1/(\sqrt{L}k^{2/3})$ (Proposition 3.4 in [Laborde and Oberman, 2020]) together with accelerated noisy gradient descent (Per-FE-C) in [Laborde and Oberman, 2020]. For NNAG we used $s_k = 1/(\sqrt{L}k^{3/4})$ and $\beta = L/10$. All the perturbation was done using *i.i.d.* Gaussian noise with unit variance, and we conducted 100 Monte-Carlo runs. The results are presented in Figure 2 right panel. As shown, NNAG outperforms all the other methods in this case.

In a related experiment, we combined NNAG with SGD and SVRG. For the SGD-mixing, we replaced the noisy gradients with SGD-like gradients, while for the SVRG-mixing we evaluated all the gradients at the beginning of each epoch (essentially like taking a snapshot in SVRG) and set $t_k = 0$. The step-sizes for SVRG (in accordance with the original implementation [Johnson and Zhang, 2013]), SGD, NNAG+SVRG, and NNAG+SGD were set as $1/(10L)$, $1/L$, $c = 1/\sqrt{L}, \beta = L/10$,

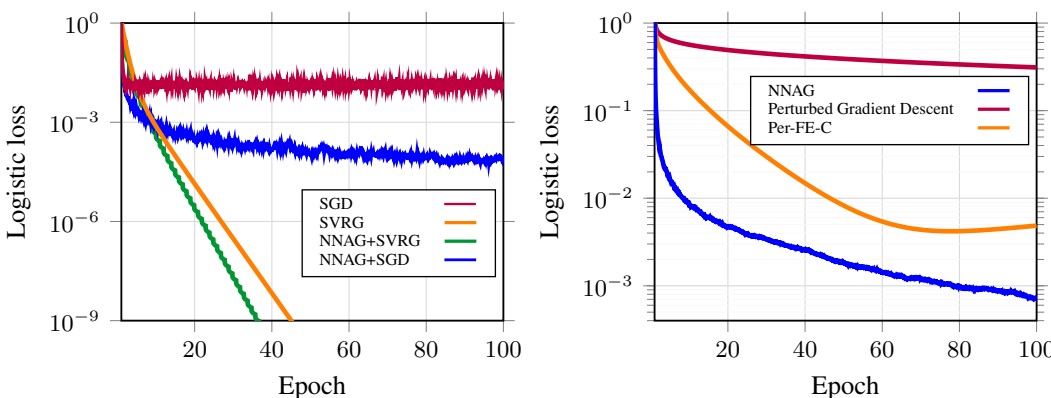

Figure 2: Comparison of the performance of various methods in binary classification problem.

and $c = 1/\sqrt{L}, \beta = L$, respectively. We conducted 100 Monte-Carlo runs, and the results are displayed in Figure 2 left panel. Notably, when mixed with either SGD or SVRG, NNAG outperforms the original methods. This highlights NNAG's flexibility in terms of gradient noise and demonstrates its potential to accelerate various methods.

**Classification on CIFAR10.** Finally, we tackled the non-convex optimization problem of training a CNN with CIFAR10 dataset [Krizhevsky et al., 2009] using the SGD, SVRG, NNAG, and NNAG+SVRG methods. The network consisted of two convolutional layers each followed by max pooling and 3 fully connected linear layers each followed by ReLU activation function. The stepsizes for SGD and SVRG were set as 0.01, and for the NNAG and NNAG+SVRG algorithms we had $c = 0.05, \beta = 150^2$ and $c = 0.001, \beta = 100^2/10$, respectively. The division by 10 is due to stepsize division by 10 in the SVRG method. The results for 20 Monte-Carlo simulations are depicted in Figure 3. Notably, SVRG+NNAG outperforms the other methods in terms of minimizing training error. Additionally, NNAG exhibits slightly better validation accuracy, hinting at its convergence toward a different local solution.

It is worth noting that faster convergence rates of NNAG+SVRG do not pose problems of overfitting. If necessary, one can terminate the algorithm earlier to achieve optimal performance. In Figure 3, for instance, NNAG+SVRG reaches its peak "validation accuracy" after approximately 20 epochs, with a validation accuracy of roughly 0.6 and a training error of around 0.66. After this point, a slow overfitting phase begins. Similarly, for SGD and SVRG, their peak "validation accuracy" is achieved after about 50 epochs, with a validation accuracy of approximately 0.6 and a training error of about 0.66, followed by a slow overfitting phase. Finally, NNAG achieves comparable results after approximately 100 epochs.

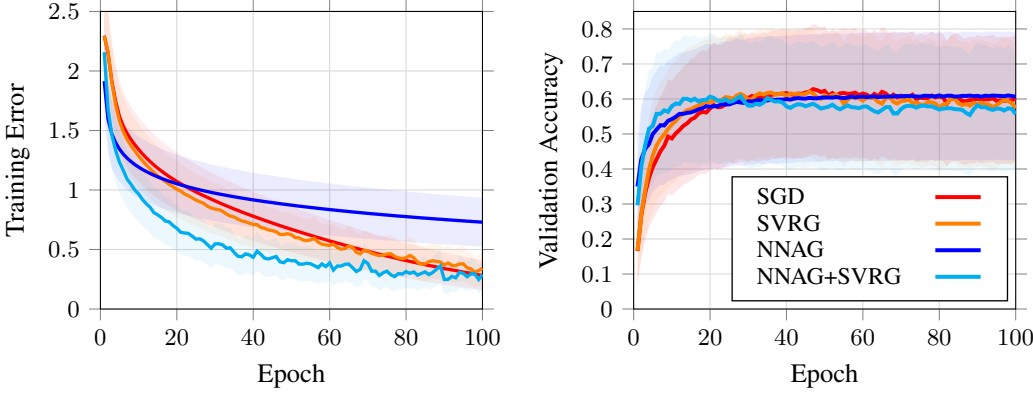

Figure 3: Training error and validation accuracy of NNAG, SGD, SVRG, and NNAG+SVRG when used for training a simple CNN on CIFAR10 dataset. Lower and upper confidence bounds with significance level of 0.68 are drawn with similar color to their corresponding line.

# 7 Related Work

Polyak's Heavy-Ball (HB) method was one of the first momentum-based methods which could accelerate relative to the gradient descent method [Polyak, 1963]. However, this was not the case for every smooth function [Lessard et al., 2016]. Nesterov modified the HB and introduced the NAG method. This method achieved global convergence with a rate of $\mathcal{O}(1/k^2)$ for smooth convex functions [Nesterov, 1983]. Nesterov used estimate sequence technique to show the convergence of the NAG method. This technique does not provide immediate insights toward the success of the NAG algorithm in acceleration. Thus, many have tried to understand the essence of acceleration.

On a similar line of work, Wibisono et al. [2016] introduced a variational perspective on accelerated methods, leading to a general (non-Euclidean) ODE containing the ODE found by Su et al. [2016] as a special case. Their approach is based on the choice of a Lagrangian and its corresponding parameters. Since the choice of Lagrangian is not unique, Wilson et al. [2021] provided a variational perspective on different accelerated first-order methods using another Lagrangian. Fazlyab et al. [2017] developed a family of accelerated dual algorithms for constrained convex minimization through a similar variational approach. In a more recent development, Zhang et al. [2021] showed that the second-variation also plays an important role in optimality of the ODE found by Su et al. [2016]. Specifically, they showed that if the time duration is long enough, then the mentioned ODE for the NAG algorithm is the saddle point to the problem of minimizing the action functional.

The dynamical system perspective on NAG was studied in [Muehlebach and Jordan, 2019]. They showed that the NAG algorithm is recovered from the SIE discretization of an ODE. The mentioned ODE was not the result of a vanishing step-size argument. They found that a curvature-dependent damping term accounts for the acceleration phenomenon. Interestingly, [Chen et al., 2022] also used similar ODE without the SIE discretization. They showed that implicit-velocity is the reason of the acceleration. In a recent analysis, [Muehlebach and Jordan, 2023] explores the connections between non-smooth dynamical systems and first-order methods for constrained optimization.

# 8 Conclusion

In this work, we tackled the problem of unconstrained smooth convex minimization in Euclidean space. Through a variational analysis of HR-ODEs, we achieved improved convergence rates for NAG in terms of gradient norm. In addition, we showed that NAG can be viewed as an approximation of the rate-matching technique when applied on a specific ODE. Our analysis was then extended to stochastic scenarios. In particular, we proposed a method with both constant and varying stepsizes which performed comparable and sometimes better than state of the art methods.

This work entails multiple future directions. Nesterov's oracle complexity lower bound on gradient norm minimization is $\mathcal{O}(k^{-4})$ [Nesterov, 2003]. It remains an open question to see if the NAG method can achieve this rate of convergence for gradient norm minimization. In this work, we noticed that the HR-ODEs follow the same external force structure. In the smooth-strongly convex case, Triple Momentum (TM) method is the fastest known globally convergent method [Van Scoy et al., 2018]. However, the HR-ODE associated with the TM method is not shown to achieve the similar convergence rate as the TM method [Sun et al., 2020]. One could use the external force structure proposed here to find a better convergence rate for the HR-ODE associated with the TM algorithm. In addition, our analysis was confined to the Euclidean space. We believe it is possible to explore non-Euclidean forces using a Bregman Lagrangian as in [Wibisono et al., 2016]. Finally, we blended our noisy stochastic scheme with other known stochastic methods (*e.g.* SGD and SVRG). This technique improved the performance of those methods. As a future work, one can apply the same technique to other practical methods like ADAM, RMSprop, etc, and study the behavior of the final algorithm.

## Acknowledgements

Alp Yurtsever and Hoomaan Maskan received support from the Wallenberg AI, Autonomous Systems and Software Program (WASP) funded by the Knut and Alice Wallenberg Foundation. Konstantinos C. Zygalakis acknowledges support from a Leverhulme Research Fellowship (RF/2020-310), and the EPSRC grant EP/V006177/1. We gratefully acknowledge the support of NVIDIA Corporation with the donation of 2 Quadro RTX 6000 GPUs used for this research.

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
