# OpenReview forum: "A Variational Perspective on High-Resolution ODEs"
_NeurIPS.cc/2023/Conference — NeurIPS 2023 poster_

### Official Review · Reviewer_rs2m · 2023-07-03

**Soundness:** 4 excellent
**Presentation:** 3 good
**Contribution:** 4 excellent
**Rating:** 7
**Confidence:** 3

**Summary:**

This paper introduces forced Lagrangians in order to better understand discrete schemes for accelerate convex numerical optimization from a continuous-time ODE perspective.

**Strengths:**

Two choices of F enable to re-derive continuous-time ODEs which were introduced recently in the literature (Section 2). Likewise, accelerated discrete schemes derive subsequently (Sections 3, 4) recover known schemes or in slightly modified form with slightly superior convergence rates. Empirically, it is demonstrated that even for non-convex network training, applying stochastic versions of accelerated schemes may perform well.

This paper significantly contributes to our understanding what unconstrained convex optimization with optimal convergence rates really means.

**Weaknesses:**

Authors use their jargon right from the beginning which makes the paper hard to read for readers who do not work on similar topics. For example, what “LR-ODE”, “HR-ODE” and the “rate matching technique” means should be briefly explained to a broader educated readership in the introduction.

**Questions:**

none

**Limitations:**

see "weaknesses" on how to improve the presentation

---

> ### Author Rebuttal · Authors · 2023-08-09
>
> Thank you for your valuable feedback and comments. We will update the introduction of our camera ready version with a “Background” section where we describe the terms like “LR-ODE”, “HR-ODE” and “rate matching technique” in order to reach a wider audience, as thoughtfully suggested by the reviewer.

---

### Official Review · Reviewer_A9z2 · 2023-07-04

**Soundness:** 4 excellent
**Presentation:** 3 good
**Contribution:** 3 good
**Rating:** 7
**Confidence:** 2

**Summary:**

This work combines variational perspective approach with high-resolution ODE functions to investigate the Nesterov accelerated gradient descent algorithm (NAG). With variational perspective, the authors reconstruct various high-resolution ODEs derived in previous research using alternative methods. Moreover, through this approach, they propose a special representation of NAG that exhibits an improved convergence rate in terms of gradient norm minimization. The authors also discuss some new properties of rate-matching technique. Finally, the authors analyze the stochastic setting both theoretically and empirically.

**Strengths:**

1.	The idea of combining variational perspective and high-resolution ODEs by including external forces is very interesting.
2.	The authors show several important theoretical results in their manuscript. By carefully checking their proofs of theorems (except for Section 5 because of time), I think overall these results are correct.
3.	The numerical results indicate the potential of having better optimization algorithms based on the theoretical results in this manuscript.
4.	Overall, the manuscript is well written.

Overall, I have a positive impression of this work, however, I also admit that I am not an expert in this specific field, which may affect my confidence in assessing its accuracy and significance.


**Weaknesses:**

1.	It would be better to have a more detailed introduction to low-resolution ODEs and high-resolution ODEs. Without reading some previous work, it is difficult to understand the differences between low-resolution ODEs and high-resolution ODEs. Are high-resolution ODEs the ODE functions that contain the learning rate $s$? Why do people need to care about high-resolution ODEs?
2.	I feel that the proof of Proposition 4.1 is more of an intuition rather than a rigorous demonstration. The treatment of the condition $s\to 0$ seems imprecise, as sometimes the authors will directly consider this condition as $s=0$ (e.g. line 60), while at other times the authors maintain $s$ to be a non-zero value (e.g. line 59). The use of the word “approximately” in Prop 4.1 is also vague.
3.	Some typos/ unclear parts in the proofs that the authors may need to double-check.

(a) line 84, $\frac{\partial L}{\partial X}(X_t,...)$ should be $\frac{\partial L}{\partial X_t}(X_t,...)$.

(b) equation (7), left hand side, $\bigtriangledown f$ should be $\bigtriangledown f(X_t)$.

(c) Theorem 2.1, $\dot{\gamma}=e^{\alpha t}$ should be $\dot{\gamma}=e^{\alpha_t}$.

(d) line 138, ODE(14) should be ODE(12).

(e) equation (25) $(3/t+\sqrt{s}\bigtriangledown f(X_t))$ should be $(3/t+\sqrt{s}\bigtriangledown^2 f(X_t))$.

(f) line 196, $\sigma$ not introduced, is it the variance of the noise?

(g) equation (38) (appendix), second line $\sqrt(s)e^{-\alpha_t}\ddot{\beta_t}$ should be $\sqrt(s)e^{-2\alpha_t}\ddot{\beta_t}$.

(h) line 392, (7) should be (11).

(i) line 406, first equality $+\frac{1}{2}||v_k-x^*||^2$ should be $-\frac{1}{2}||v_k-x^*||^2$.

(j) line 408, the term $+\frac{s^2(k+2)}{4}||\bigtriangledown f(x_{k+1})||^2$ in the second inequality is left out, therefore the authors need to double-check whether the results still hold after considering this term.

(k) line 411, the first equaltiy redundant,

(l) line 413 "not ethat" typo

(m) line 421, the term $\frac{ks}{2}(\bigtriangledown f(x_k-\bigtriangledown f(x_k))$ miss ")".

(n) equation (62), $(3/t+\sqrt{s}\bigtriangledown f(X_t))$ should be $(3/t+\sqrt{s}\bigtriangledown^2 f(X_t))$.

4. Figure 1 is hard to read. (font size too small)
5. It would be better to describe what "NAG" means (Nesterov accelerated gradient?) the first time this abbreviation is used.


**Questions:**

1.	It is unclear to me that equation (21) and equation (27) (by replacing one term) is equal to NAG (line 76), why is that?
2.	In line 211, the authors say that practically $k_0$ is lower than the term $(\cdot)^{1/\alpha}$. However, one of the conditions in Theorem 5.1 and Theorem 5.2 is that $k_0\geq (\cdot)^{1/\alpha}$, does that mean this condition will not be satisfied in practice?


**Limitations:**

The authors don't specifically discuss the limitations of this work. The authors may consider adding a paragraph in their manuscript to discuss the limitations of their work based on the reviewer's feedback.

I don't think there will be a significant negative societal impact of this work.

---

> ### Author Rebuttal · Authors · 2023-08-09
>
> Thank you for your detailed and valuable comments. We are pleased that you recognized the strength of our paper in terms of novelty, significance, soundness, and presentation. Below, you can find our responses addressing the concerns you raised.
>
> **Reviewer**: “It would be better to have a more detailed introduction to low-resolution ODEs and high-resolution ODEs. Without reading some previous work, it is difficult to understand the differences between low-resolution ODEs and high-resolution ODEs. Are high-resolution ODEs the ODE functions that contain the learning rates? Why do people need to care about high-resolution ODEs?”
>
> **Authors**: We will update the introduction of our camera ready version with a “Background” section where we describe the terms like “LR-ODE”, “HR-ODE” and “rate matching technique” and their significance to our analysis in order to reach a wider audience, as thoughtfully suggested by the reviewer.
>
> The HR-ODEs contains the step-size (learning rate) ‘s’ allowing them to effectively incorporate Nesterov's Accelerated Gradient (NAG) method as the step-size changes. In contrast, this is not the case for LR-ODEs which were independent of the step-size (for an illustrative description, refer to [3, Figure 2]).
>
> The importance of HR-ODEs lies in their direct connection to accelerated algorithms. These ODEs can be discretized using the common discretization techniques like the semi-implicit Euler discretizer and recover well-known methods like the NAG algorithm. This is not the case for LR-ODEs in [1], where more complicated and less intuitive discretizers (like the rate-matching technique in [2]) are needed.
>
> **Reviewer**: “I feel that the proof of Proposition 4.1 is more of an intuition rather than a rigorous demonstration. The treatment of the condition $s\rightarrow 0$ seems imprecise, as sometimes the authors will directly consider this condition as $s=0$ (e.g. line 60), while at other times the authors maintain s to be a non-zero value (e.g. line 59). The use of the word “approximately” in Prop 4.1 is also vague.”
>
> **Authors**: We use the term 'approximation' because we employ approximations as in equations (59) and (60). The reason for utilizing different forms of approximations is as follows: Given that $s$ converges to zero more rapidly than $\sqrt{s},$ we neglect $s$. This same 'approximation' strategy is used in [3, equations (2.2),(2.3)]. We will provide a clear explanation of the concept of 'approximation' in this specific context in our camera-ready version.
>
> **Reviewer**: “Some typos/ unclear parts in the proofs that the authors may need to double-check.”
> **Authors**: Thank you for your careful reading and detailed comments. We will fix these typos and mistakes in our camera ready version. Particularly,
> (f): Yes, $\sigma^2$ is the noise variance.
> (j): Thanks for noticing this, it is easy to fix this problem. We simply forgot updating the coefficient of $||\nabla f(x_{k+1})||^2$   to $\frac{s(k+2)}{4}\left(\frac{1}{L}-s\right)$. This means that the term you pointed out should be combined with the second $||\nabla f(x_{k+1})||^2$. Therefore, the last 2 lines of (48) should be
> $$=-\frac{s(k+2)k}{8}\left(\frac{1}{L}-s\right)||\nabla f(x_{k+1}-\nabla f(x_k))||^2-\frac{s(k+2)}{4}\left(\frac{1}{L}-s\right)||\nabla f(x_{k+1}||^2-\frac{s^2(k+2)k}{8}||\nabla f(x_{k})||^2$$
> $$\leq -\frac{s^2(k+2)k}{8}||\nabla f(x_{k})||^2$$
>
> **Reviewer**: “Figure 1 is hard to read. (font size too small)”
> **Authors**:  We will make the font size larger for the camera-ready version.
>
> **Reviewer**:  “It would be better to describe what "NAG" means (Nesterov accelerated gradient?) the first time this abbreviation is used.”
> **Authors**: We will clarify this in our camera ready version.
>
> **Question 1**: “It is unclear to me that equation (21) and equation (27) (by replacing one term) is equal to NAG (line 76), why is that?”
>
> **Authors**: This can be seen when one writes down the one line representation of the update (21) or (27) with one term replacement. The pathway is as follows: consider (21), write $v_k$ as a function of $x_k,x_{k+1}$ through the first line of the update (21). Then, replace in the second line and rearrange the terms and get
> $$x_{k+2}=x_{k+1}+\frac{k}{k+3}(x_{k+1}-x_k)-\frac{sk}{k+3}\left( \nabla f(x_{k+1}-\nabla f(x_k)\right)-s\nabla f(x_{k+1}) $$
>
> which is the one-line representation of the NAG method. For (27) after replacing that one term, same approach for the sequence $x_k$ (eliminating $y_k$'s and $v_k$'s) leads to the one-line representation of the NAG method.
>
> **Question 2**: “In line 211, the authors say that practically $k_0$ is lower than the term $(⋅)^{1/α}$. However, one of the conditions in Theorem 5.1 and Theorem 5.2 is that $k_0 \geq (⋅)^{1/α}$, does that mean this condition will not be satisfied in practice?”
>
> **Authors**: The condition on $k_0$ is the minimum number of iterations needed for our theoretical guarantees to hold. In line 211 we mean that in practice the method achieves the error bounds in Theorems 5.1 and 5.2 even for the smaller number of iterations than the theoretical bound $(⋅)^{1/α}$. In a sense, the method works better than the theory suggests.
>
> Thank you for your careful reading. We will clarify this statement to avoid confusions.
>
> **Limitations**: “The authors may consider adding a paragraph in their manuscript to discuss the limitations of their work based on the reviewer's feedback.”
>
> **Authors**: We will add this paragraph, thank you for your suggestions.
>
> **References**
>
> [1] W. Su, S. Boyd, E. J. Candes, "A Dfferential Equation for Modeling Nesterov's Accelerated Gradient Method: Theory and Insights" (2016)
>
> [2] A. Wibisono, A. C. Wilson, and M. I. Jordan, "A variational perspective on accelerated methods in optimization" (2016)
>
> [3] B. Shi, S. S. Du, M. I. Jordan, and W. J. Su. "Understanding the acceleration phenomenon via high-resolution differential equations" (2021).

---

> > ### Comment · Reviewer_A9z2 · 2023-08-13
> > **Thank you for the response**
> >
> > I would like to thank the authors for their response. I have one follow up question. I agree that equation (21) can be transferred to the form of the one-line representation above, However, according to line 76, the one-line representation of NAG should be $$x_{k+2}=x_{k+1}+\frac{k+1}{k+4}(x_{k+1}-x_{k})-\frac{s(k+1)}{k+4}(\bigtriangledown f(x_{k+1})-\bigtriangledown f(x_{k}))-s\bigtriangledown f(x_{k+1})$$. Do these differences matter?

---

> > > ### Author Response · Authors · 2023-08-15
> > > **Response 1**
> > >
> > > Thank you for your comments. The relation of the NAG (the one-line update) holds for a 3-point sequence, it does not matter if the sequence is $x_k,x_{k+1},x_{k+2}$ or $x_{k-1},x_{k},x_{k+1}$. This is just a matter of notation/convention; in fact, we can define a new sequence $z_k:=x_{k+1}$ and get exactly the same one-line update as the NAG.

---

> > > > ### Comment · Reviewer_A9z2 · 2023-08-15
> > > >
> > > > Thank you for your response. The difference I notice here is not whether the sequence is represented by $x_k,x_{k+1},x_{k+2}$ or $x_{k-1},x_{k},x_{k+1}$, the difference is the coefficient $(k+1)/(k+4)$ vs. $k/(k+3)$ given the same representation, this difference cannot be removed by a simple transformation $z_{k}:=x_{k+1}$ (because then the coefficients containing $k$ would also need to be changed). Although I guess this difference should be fine?

---

> > > > > ### Author Response · Authors · 2023-08-15
> > > > >
> > > > > Thanks for your comment. The equation in line 76 is equivalent to
> > > > > $$x_{k+1}=x_k+\frac{k}{k+3}(x_k-x_{k-1})-\frac{s(k)}{k+3}(\nabla f(x_k)-\nabla f(x_{k-1}))-s\nabla f(x_k)$$
> > > > > with an initial point $x_{-1}$.
> > > > >
> > > > > And the one-line representation of (21) is
> > > > > $$\hat x_{k+2}= \hat x_{k+1}+\frac{k}{k+3}(\hat x_{k+1}-\hat x_{k})-\frac{s(k)}{k+3}(\nabla f(\hat x_{k+1})-\nabla f(\hat x_{k}))-s\nabla f(\hat x_{k+1})$$
> > > > > with an initial point $\hat x_0$.
> > > > >
> > > > > Now, if we introduce $z_k:=\hat x_{k+1}$, we get
> > > > > $$z_{k+1}=z_k+\frac{k}{k+3}(z_k-z_{k-1})-\frac{s(k)}{k+3}(\nabla f(z_k)-\nabla f(z_{k-1}))-s\nabla f(z_k),$$
> > > > > with initial point $z_{-1}$; which indicates that the updates for the sequence $z_k$ are exactly the same as the ones for sequence $x_k$.

---

> > > > > > ### Comment · Reviewer_A9z2 · 2023-08-15
> > > > > >
> > > > > > Thanks, I got this point. I decide to increase the score by one based on the responses from the reviewers.

---

### Official Review · Reviewer_ZUpb · 2023-07-04

**Soundness:** 2 fair
**Presentation:** 2 fair
**Contribution:** 2 fair
**Rating:** 6
**Confidence:** 1

**Summary:**

They generalize the Lagrangian formulation of known first order optimization methods [Wibisono et al., 2016, Wilson et al., 2021] by introducing the notion of external forces, and show theorems on convergence of convex/strongly convex functions.

In addition, they show the variational analysis leads a special representation of NAG, and it enjoys to superior convergence rates than [Shi et al., 2019]. Based on the special representation of NAG, they generalize it to stochastic variation in Section 5, called NNAG, and show its convergent theorem.

They check the practical behavior of the proposed NNAG by using
- binary classification,
- classiﬁcation on CIFAR10.

**Strengths:**

originality
- Introducing the external force term to the Lagrangian formulation and show convergent theorems with it.

quality/clarity
- They show their idea and mathematical statements clearly, and show their proofs.

significance
- Their formulation can lead known dynamics also. In this sense, their proposal can be regarded as a unification of various optimization dynamics.

**Weaknesses:**

- The first part of the paper sounds natural, but to me, there seems no theoretical reason to introduce i.i.d. noise to the gradients in section 5.
- In my opinion, they should conduct more numerical experiments. For example, the experiments in classification on CIFAR10, NNAG and SVRG+NNAG are competitive to SGD. This result, itself is good, but it means there is no big incentive to use the proposed algorithm, but SGD.

**Questions:**

- Is there any natural/theoretical motivation to introduce i.i.d. noise to the gradients in section 5?
- Is there any strong incentive to use NNAG or its variants compared to SGD in practice?
- Besides it, the authors wrote `As the figure depicts, the SVRG+NNAG performs faster than the other methods in terms of minimizing the training error.`, but minimizing training error itself sounds possibility of overfitting in machine learning context. If not, please correct me.

---

> ### Author Rebuttal · Authors · 2023-08-09
>
> Thank you for your valuable feedback and comments. We are pleased that you recognized the strength of our paper in terms of originality, quality/clarity, and significance. Below, you will find our responses addressing the concerns you raised.
>
> **Reviewer**: “The first part of the paper sounds natural, but to me, there seems no theoretical reason to introduce i.i.d. noise to the gradients in section 5. Is there any natural/theoretical motivation to introduce i.i.d. noise to the gradients in section 5?”
>
> **Authors**: The motivation for studying the model with i.i.d. noise in Section 5 is twofold: From a practical standpoint, this choice is motivated by the fact that many algorithms employed in machine learning (for example for training neural networks) are characterized by noisy gradients [1,2,3]. Thus, adding noise to the gradients takes our theoretical results from the earlier sections one step closer to the real-world practical scenarios. In addition, from a theoretical standpoint, our interest lay in probing the novel representation (21). For instance,  [1] extends the continuous time analysis of deterministic NAG from [4] to a stochastic setting. Similarly, we extended our analysis in Section 5 to investigate the impacts of the new representation (21) in the stochastic setting.
>
> **Reviewer**: “In my opinion, they should conduct more numerical experiments. For example, the experiments in classification on CIFAR10, NNAG and SVRG+NNAG are competitive to SGD. This result, itself is good, but it means there is no big incentive to use the proposed algorithm, but SGD. Is there any strong incentive to use NNAG or its variants compared to SGD in practice?”
>
> **Authors**: As an incentive, we can highlight that SVRG+NNAG outperforms SGD in terms of convergence rate in our experiments. However, we wish to emphasize that our primary contribution lies in a new comprehension of acceleration, and SVRG+NNAG simply serves as a proof-of-concept resulting from this insight.
>
> The main contribution of our paper is a novel understanding of the acceleration phenomenon through an innovative extension on the continuous time analysis of the Nesterov’s accelerated gradient (NAG) method that leverages the forced Euler-Lagrange equation that we present in Section 2. Although NAG itself is mathematically well-founded, the particular mechanisms behind its effectiveness are not immediately obvious and often considered ‘mysterious’; please see lines 28-50 in our paper for a short survey on different attempts to ‘demystify’ this phenomenon. In this context, our novel understanding of acceleration through the forced Euler-Lagrange equation is interesting in its own right. Nevertheless, beyond its theoretical significance, we believe that this novel perspective also holds great potential for deriving new results and practical algorithms. We mention some immediate implications in Sections 3, 4, and 5, with SVRG+NNAG being just one of these examples. Our goal in Section 5 is to demonstrate the potential of exploring different representations (like (21)) in practice. This idea is further emphasized in the future directions, including the combination of NNAG with ADAM or RMSprop.
>
> Finally, we are open to the idea of including more numerical experiments if the reviewer can provide specific recommendations regarding the suggested experiments.
>
> **Reviewer**: “Besides it, the authors wrote "...", but minimizing training error itself sounds possibility of overfitting in machine learning context. If not, please correct me.”
>
> **Authors**: In the context of neural networks (or non-convex optimization in general), there is always the possibility of overfitting as well as converging to a poor local minimum that does not generalize well. This is why we include both "validation accuracy" and "training error" plots in Figure 2. These plots help us ensure that these issues do not arise in our experiments.
>
> It is important to note that faster convergence rates of NNAG+SVRG do not pose problems of overfitting. If needed, one can terminate the algorithm earlier to achieve top performance more quickly. This observation is evident in Figure 2. For instance, NNAG+SVRG achieves the top "validation accuracy" after around 20 epochs (with a validation accuracy of approximately 0.6 and a training error of approximately 0.66), after which a slow overfitting phase begins. Similarly, for SGD and SVRG, their peak "validation accuracy" results are achieved after approximately 50 epochs (with a validation accuracy of around 0.6 and a training error of about 0.66), followed by an overfitting trend. Finally, NNAG achieves comparable results after roughly 100 epochs. We will include this discussion in our camera ready version.
>
> Finally, please note that any technical analysis of the generalization error and/or overfitting is outside of the scope of our paper.
>
>
> **Before we conclude**, we would like to kindly express our surprise regarding your relatively low score for our paper, especially because your review acknowledges the strengths of our paper in terms of originality, quality/clarity, and significance. Based on your feedback and statements, it appears (in our opinion) that the positive aspects of our work are more crowded than the negative ones. We have noticed that your concerns are focused on the stochastic extension in Section 5. In light of the comprehensive responses we have provided above for your concerns, we hope that you would consider reevaluating your score for our submission.
>
> **References**
>
> [1] M. Laborde and A. Oberman. “A Lyapunov analysis for accelerated gradient methods: from deterministic to stochastic case” (2020)
>
> [2] A. Defazio, F. Bach, S. Lacoste-Julien, "SAGA: A Fast Incremental Gradient Method With Support for Non-Strongly Convex Composite Objectives" (2014)
>
> [3] J. Wu et al. “On the Noisy Gradient Descent that Generalizes as SGD” (2019)

---

> > ### Comment · Reviewer_ZUpb · 2023-08-16
> >
> > I would like to thank the authors for their response. Their response to my questions:
> > - motivation to introduce i.i.d. noise
> >     - The authors answered that it is a one step closer to the real-world practical scenarios like SGD in training of neural network. This makes sense from practical perspective.
> > - incentive to use NNAG
> >     - The authors answered that "SVRG+NNAG outperforms SGD in terms of convergence rate in our experiments". In addition, I understand that the contribution of this paper is more theoretical, a new interpretation of the NAG method by the forced Euler-Lagrange equation.
> > - on overfitting problem
> >     - The author explained why "faster convergence rates of NNAG+SVRG do not pose problems of overfitting" in the rebuttal, and I agree with the author's point.
> >
> > In summary, I can say that the author's explanation has, for the most part, addressed my concerns. So I would like to raise my score  to weak accept.

---

### Official Review · Reviewer_CWtz · 2023-08-01

**Soundness:** 3 good
**Presentation:** 2 fair
**Contribution:** 3 good
**Rating:** 7
**Confidence:** 2

**Summary:**

This paper did four works about High-Resolution ODEs and first-order optimization algorithms. The first part uses forced Euler-Lagrange equations to generalize the analysis of Low-Resolution ODEs to High-Resolution ODEs. The second part is a refined result for bound estimation of gradient norm. The third part is an interpretation of Nesterov’s acceleration by rate-matching discretization. In the last part, the authors propose a stochastic version of Nesterov’s accelerated gradient method, named NNAG in this paper, and compared it to several known stochastic methods on binary classification and training CNN.

**Strengths:**

This paper proposes a new idea of external force and uses the idea of forced Euler-Lagrange equations to generalize the analysis of Low-Resolution ODEs to High-Resolution ODEs. This leads to a new formulation of NAG. Combined with rate-match discretization, the authors discover a new connection of NAG and the continuous ODE, i.e., it is understood as a perturbation of the Low-Resolution ODE. The new formulation of NAG further inspires the framework of Noisy NAG. The idea of forced Euler-Lagrange equations may bring new insight to fast algorithm design.

**Weaknesses:**

Although this paper provides a new reformulation of NAG, as the author suggested, the refined convergence rate of gradient norm for NAG is already appeared in [S. Chen, B. Shi, and Y.-x. Yuan. Gradient norm minimization of nesterov acceleration: o(1/k^3). arXiv preprint arXiv:2209.08862, 2022]. The Lyapunov function is essentially the same using the implicit-velocity form and the explicit-velocity form in [B. Shi, S. S. Du, M. I. Jordan, and W. J. Su. Understanding the acceleration phenomenon via high-resolution differential equations. ArXiv, abs/1810.08907, 2021].

**Questions:**

The Lagrange function is kinetic energy minus potential energy (though multiplied by some decreasing coefficient). The Lagrange function and external force is considered separately in this paper. Is it possible to give physical explanation about the acceleration by forced Euler-Lagrange equation and use different energy function and external force to inspire better algorithm design?

**Limitations:**

The authors have adequately addressed the limitations and have indicated problems for further study. This work appears to have little negative societal impact.

---

> ### Author Rebuttal · Authors · 2023-08-09
>
> Thank you for your valuable feedback and comments. In what follows, we address and respond the points raised by the reviewer.
>
>
> **Reviewer**: "Although this paper provides a new reformulation of NAG, as the author suggested, the refined convergence rate of gradient norm for NAG is already appeared in [Chen et al., 2022]."
>
> **Authors**: [Chen et al., 2022] appeared while we were already working on our paper. Although they achieve essentially the same convergence rate, the analysis techniques are significantly different. In particular, they follow an implicit velocity perspective leading to a different Lyapunov analysis, whereas we introduced the forced Euler-Lagrange perspective. Despite the end results being similar, we believe that our analysis is interesting in its own right. In particular, the use of external forces can inspire new methods or new convergence results for existing methods.
>
>
> **Reviewer**: "The Lyapunov function is essentially the same using the implicit-velocity form and the explicit-velocity form in [She et al., 2021]."
>
> **Authors**: Although our Lyapunov function is essentially the same as in [1] and [2], we would like to highlight that our new representation (please see (21)) enhances the understanding behind the selection of this Lyapunov function. This is due to the fact that the second component of this Lyapunov function, represented as
> $$\left( \frac{1}{2}||x_{k+1}-x^*+\frac{k}{2}(x_{k+1}-x_k)+\frac{ks}{2}\nabla f(x_k)||^2 \right)$$
> equates to
> $$\frac{1}{2}||v_k-x^*||^2$$
> where $v_k$ is defined in (21) (it is easy to see this simply by rewriting $v_k$ as a function of $x_k$ and $x_{k+1}$ by using the first line of (21)). This intuitive understanding seems to be missing in the prior work.
>
> In addition, please note that one of the main goals of [1] is to simplify the analysis in [2]. Our work significantly advances this aim: Upon comparing the proofs of Theorem 3.1 in [1], Theorem 6 in [2], and Theorem 3.1 in our work (see Appendix A.4), it is apparent that our new representation substantially simplifies the algebraic aspects of the proof.
>
> We will further clarify these connections and make comparisons with the existing approaches in the camera-ready version.
>
>
> **Reviewer**: "Is it possible to give physical explanation about the acceleration by forced Euler-Lagrange equation and use different energy function and external force to inspire better algorithm design?"
>
> **Authors**: From a physical perspective, the forces are non-conservative (line 87 in our paper) meaning that they are dissipative in nature, similar to friction or air resistance. This characterization provides a more comprehensive and accurate formulation compared to that in [3], which depicts a particle losing potential energy and gaining kinetic energy and therefore accelerating. In this respect, proper choice of the force leads to a more accurate physical model and thus better algorithm design and can even yield other interesting findings. For example, in the strongly convex regime, choosing an appropriate force to derive the HR-ODE of the TM method could unveil new convergence rates for both the HR-ODE and the algorithm (line 287 in our paper). As it stands, the HR-ODE in [4] for the TM method has a proven convergence rate that is slower than the algorithm itself. We will provide a detailed discussion on this in the camera-ready version.
>
> **References**
>
> [1] S. Chen, B. Shi, and Y. Yuan. "Gradient norm minimization of nesterov acceleration: $o(1/k^3)$" (2022)
>
> [2] B. Shi, S. S. Du, M. I. Jordan, and W. J. Su. "Understanding the acceleration phenomenon via high-resolution differential equations"(2021)
>
> [3] A. Wibisono, A. C. Wilson, and M. I. Jordan, "A variational perspective on accelerated methods in optimization" (2016)
>
> [4] B. Sun, J. George, and S. S. Kia. "High-resolution modeling of the fastest first-order optimization method for strongly convex functions" (2020)

---

> > ### Comment · Reviewer_CWtz · 2023-08-20
> >
> > I would like to thank for the author's reply. I agree that the forced Euler-Lagrangian is anothor way to help simplifying the proof in [B. Shi, S. S. Du, M. I. Jordan, and W. J. Su. "Understanding the acceleration phenomenon via high-resolution differential equations"(2021)]. Although the level of simplification compared to [S. Chen, B. Shi, and Y.-x. Yuan. "Gradient norm minimization of nesterov acceleration: o(1/k^3)"(2021)] is still not very clear to me, the authors have addressed my question. On the other hand, the physical explanation of the non-potential external force and its potential usage seem interesting to me. In summary, I decide to raise my score by one based on the responses from the authors.

---

### Decision · Program_Chairs · 2023-09-21

**Decision:**

Accept (poster)

**Comment:**

This paper studied High-Resolution ODEs and first-order optimization algorithms by giving 1) a forced Euler-Lagrange equations to generalize the analysis of Low-Resolution ODEs to High-Resolution ODEs, 2) a refined result for bound estimation of gradient norm, 3) an interpretation of Nesterov’s acceleration by rate-matching discretization, and 4) a stochastic version of Nesterov’s accelerated gradient method. All reviewers believe this paper makes an important contribution to the community. The AC agrees and recommends acceptance.